# Monitoring of Cd and GSH contents and *Bn-OASTL* expression in transgenic tobacco seedlings in response to Cd stress

**Xiaolan He**[1], **Jianwei Wang**[2]*, **Wenxu Li**[3], **Xinhong Chen**[4]

1 School of Life and Health Science, Kaili University, Kaili, GuiZhou,China, 2 School of Science, Kaili University, Kaili, GuiZhou, China, 3 Institute for Wheat Research, Henan Academy of Agricultural Sciences, Zhengzhou, Henan, China, 4 Shaanxi Key Laboratory of Genetic Engineering for Plant Breeding, College of Agronomy, Northwest A&F University, Yangling, Shaanxi, China

* agan1982@126.com

## Abstract

Cadmium (Cd) pollution threatens agricultural productivity and food safety. *O-acetylserine(thiol)lyase* (*OASTL*) genes have been tied to plant responses to heavy metal stress, yet their roles in heterologous systems, particularly in Cd accumulation and tolerance, remain unclear. Here, we isolated a novel *OASTL* gene, *BnaOASTL*, from the high-Cd-accumulating oilseed rape cultivar *Brassica napus* "Nanyou 868" and expressed it in tobacco (*Nicotiana benthamiana*). Transgenic lines were exposed to Cd stress, and Cd content, glutathione (GSH) level, and *BnaOASTL* expression were evaluated. The full-length *BnaOASTL* cDNA (969 bp) encoded a cytoplasmic/nuclear protein of 322 amino acids. Under Cd stress, *Bn-OASTL* expression was significantly upregulated in transgenic plants, particularly in roots. However, compared with wild-type, transgenic lines showed no improvement in Cd tolerance or accumulation and no significant changes in GSH levels. The findings suggest that although *BnaOASTL* is transcriptionally responsive to Cd stress, its overexpression alone does not confer altered Cd tolerance or accumulation in tobacco. The study highlights the complexity of Cd response mechanisms and suggests that *BnaOASTL* functions within a broader, species-specific regulatory network.

## 1. Introduction

Cadmium (Cd) contamination in farmland soils is a pressing environmental problem, threatening crop production and human health due to its toxicity and high mobility in plants [1,2]. Mitigation strategies such as phytoremediation and the development of Cd-safe crops depend on understanding the molecular pathways governing Cd uptake, transport, and detoxification.

The *O-acetylserine(thiol)lyase (OASTL)* family is central to cysteine (Cys) biosynthesis, which underpins production of glutathione (GSH), a key metabolite in

**Data availability statement:** All relevant data are within the paper and its Supporting Information files.

**Funding:** This research was supported by the Specialized Fund for the Doctoral Development of Kaili University (Grant No. BSFZ202503). The funders had no role in study design, data collection and analysis, decision to publish, or preparation of the manuscript.

**Competing interests:** The authors have declared that no competing interests exist.

heavy metal detoxification [3–5]. In *Arabidopsis thaliana*, depletion of *AtOASTL-A1* reduces intracellular Cys and GSH contents, increasing Cd sensitivity [6]. Since the first *OASTL* was identified in *A. thaliana* [7–10], homologs have been isolated from spinach [11], *Brassica juncea* L. [12], vetch (*Vicia sativa* L.) [13], *Glycine max* (L.) [4], *Leucaena leucocephala* [14], *Sorghum bicolor* [3], *Solanum lycopersicum* L. [15], and *Cardamine hupingshanensis* [16]. Functional analyses have confirmed that some *OASTL* genes enhance heavy metal resistance when overexpressed in transgenic plants [4].

OASTL exists in multiple isoforms localized to the cytoplasm, mitochondria, and chloroplasts [5,17]. These proteins are highly conserved across species, particularly in the diphosphate (PLP)-binding site (PXXSVKDR), substrate-binding site (TSGNT), and serine acetyltransferase (SAT) interaction site (KPGPHK) [15,18–20], underscoring their essential metabolic role. Their involvement in Cd stress is well documented. For example, overexpression of *GmOASTL4* in tobacco enhances Cd tolerance [4]. Conversely, disruption of *AtOASTL-A1*, a key factor in the final step in Cys biosynthesis [9], reduces GSH levels, increases oxidative stress, and heightens Cd sensitivity [5,6,21], highlighting the link between OASTL activity and Cd detoxification.

Despite extensive studies in other species, *OASTL* genes from high-Cd-accumulating *Brassica napus* genotypes remain poorly characterized. Recent work examined *OASTL* in selenium metabolism in *Cardamine hupingshanensis* [16] and in tomato under heavy metal stress [3], but the function of *OASTL* from Cd-accumulating rapeseed cultivars and its role in heterologous systems has not been thoroughly investigated. It is not yet known whether overexpression can confer Cd tolerance or alter accumulation patterns in a heterologous system. Here, we cloned the *BnaOASTL* gene from the high-Cd-accumulating oilseed rape cultivar "Nanyou 868" and evaluated its role in Cd stress response through heterologous expression in tobacco. This work provides functional insight into *BnaOASTL* in a heterologous context and contributes to understanding the molecular basis of Cd response mechanisms, with potential implications for developing Cd-tolerant crops via molecular breeding.

## 2. Materials and methods

### 2.1. Plants and treatment

The high Cd-accumulating *B. napus* cultivar 'Nanyou 868' ($2n = 4x = 38$, AACC) [22] was used as the gene source. *Nicotiana benthamiana* was selected for transgenic plant (TP) development. Surface-sterilized seeds were placed on half-strength Murashige and Skoog (MS) medium for germination at 25°C under a 12 h/12 h light/dark cycle in a greenhouse. For Cd treatment, five-leaf-stage seedlings were exposed to 5 mg·L$^{-1}$ CdCl$_2$·5H$_2$O for 72 h, after which leaves were harvested, snap-frozen in liquid nitrogen, and kept at −80°C.

### 2.2. Cloning of *BnaOASTL* cDNA

Total RNA was extracted using the RNeasy Plant Mini Kit (Qiagen, Germany, Cat. No. 74904) and converted into cDNA using SuperScript III Reverse Transcriptase (Invitrogen USA, Cat. No. 18080093). Two *OASTL* fragments were amplified with primers

designed specifically for *OASTL* sequence (accession no. GQ996586.1), namely, N6-OASTL-1 (DPsen): cagtGGTCTCa-caacatggcatctcgaattgctaaag, N6-OASTL-1 (DPantisen): cagtGGTCTCaattccatacagcttaacgttag; N6-OASTL-2 (DPsen): cagtGGTCTCagaatggagccaattgaaagtg, and N6-OASTL-2 (DPantisen): cagtGGTCTCatacaagcctggaaggtcattgattc using KOD FX Neo (Toyobo, Japan, Cat. No. KFX-201) on a T100 Thermal Cycler (Bio-Rad, USA). Each reaction contained 1.0 µL cDNA (50 ng), 1.0 µL each primer (10 µM), 5 µL buffer, 10 µL dNTP mix, 1 µL KOD, and 31 µL ddH$_2$O. Amplification conditions were 94°C for 5 min; 30 cycles of 30 s at 94°C, 45 s at 55°C, and 58 s at 72°C, and 10 min at 72°C. The products were purified and cloned into a custom vector via the "Golden Gate" method and subsequently sequenced [23].

## 2.3. Identification and characterization of *BnaOASTL*

The open reading frame (ORF) of *BnaOASTL* was identified using NCBI ORF Finder (https://www.ncbi.nlm.nih.gov/orffinder/). Subcellular localization was predicted with PSORT (http://psort.ims.u-tokyo.ac.jp/). The isoelectric point (pI) and molecular weight (MW), and other physicochemical properties were obtained using the Expert Protein Analysis System (ExPASy) (http://cn.expasy.org/). Multiple sequences were aligned using ClustalX and visualized in GenDoc. A phylogenetic tree was generated in MEGA7.0 utilizing the neighbor-joining (NJ) method.

## 2.4. Vector construction and tobacco transformation

The *BnaOASTL* ORF was cloned into vector pBWA(V)HS-ccdb-osgfp (BioRun, Wuhan, China) via Golden Gate assembly (Fig 1). Sequence-verified plasmids were introduced into *Agrobacterium tumefaciens* strain EHA105. Transgenic tobacco plants were established via the leaf-disc method and screened on MS medium with 20 mg/L hygromycin B.

## 2.5. Subcellular localization

pBWA(V)HS-*BnaOASTL*-osgfp and empty control vectors were introduced into *A. tumefaciens* strain GV3101 and transiently expressed in *N. benthamiana* leaves via agroinfiltration. GFP signals were observed 48 h post-infiltration under a confocal microscope (OLYMPUS IX71, Japan) at 488 nm excitation and 510 nm emission.

## 2.6. Molecular characterization of transgenic plants

DNA from T$_3$ transgenic and control plants (untransformed and vector control) was extracted by the CTAB method and screened for *BnaOASTL* transgene using PCR with primers FP (5′-ttcatttggagagaacacgggggac-3′) and RP (5′-gttctcaaactgttggagcatg-3′) in triplicate with a 58 °C annealing temperature. Products were resolved on a 1.0% agarose gel, yielding a~500 bp fragment from the positive plants.

## 2.7. Cd tolerance in transgenic plants

Cd stress was applied following [24] with slight modifications. T$_3$ transgenic and wild-type (WT) tobacco seeds were germinated on 1/2 MS medium at 25°C with a 16 h/8 h light/ dark cycle for 10 days. Five-leaf-stage T$_3$ seedlings were transferred to 9-cm pots containing perlite and vermiculite (1:1) and treated with 5 mg· kg$^{-1}$ Cd (CdCl$_2$·2.5 H$_2$O) in basal nutrient solution (pH 5.5) or 400 ml distilled water. Each treatment included three replicates, with one plant per pot, and seedling growth was monitored for one week.

## 2.8. Measurement of Cd and GCH content

After 20 days of Cd exposure, roots were immersed in 20 mmol·L$^{-1}$ Na$_2$-EDTA for 30 min, then washed three times with deionized water to remove surface-bound Cd. Samples were then separated into roots and shoots. Half of each sample was immediately frozen in liquid nitrogen for fresh tissue analysis, while the remainder was oven-dried at 75°C for Cd determination. Cd content was measured following the Chinese National Standard HJ786–2016 in China [25,26]. Dried

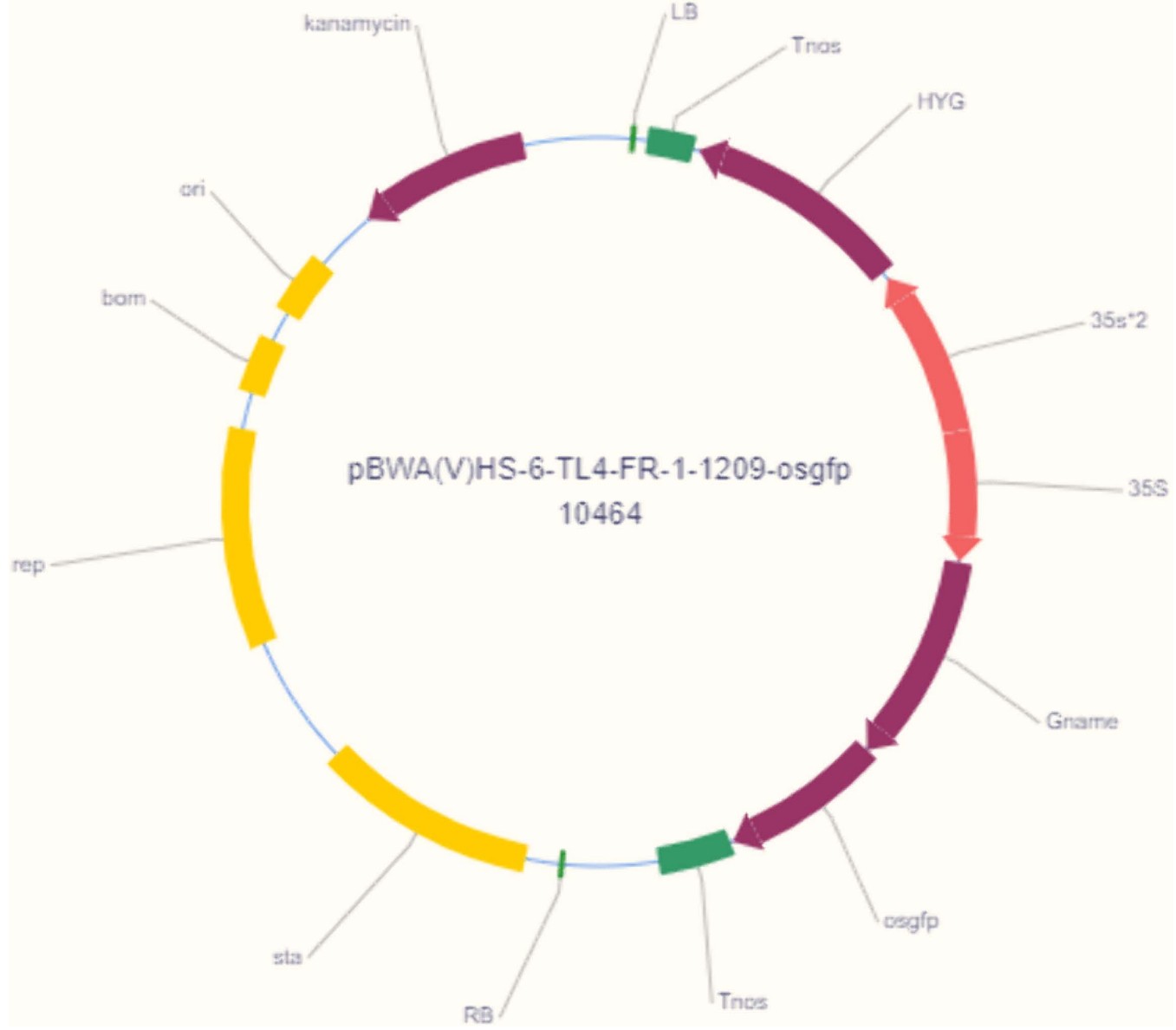

**Fig 1. Schematic of the recombinant vector.** RB and LB, right and left T-DNA borders; 35S, CaMV 35S poly A; HYG, *hygromycin B*; osgfp, the *BnaOASTL-GFP* fusion gene; Tnos, nopaline synthase terminator.

roots and shoots were weighed, pulverized, decomposed at 550°C for 8 h, and digested in 30% $HNO_3$. Cd concentrations were measured using flame atomic absorption spectrometry (Shimadzu AA-6300, Japan). Data represent the mean of three replicates. GSH was extracted from shoots of Cd-treated and control plants and quantified using a reduced glutathione (GSH) kit (Visible light spectrophotometer, Shanghai Jinghua 721).

## 2.9. Quantitative real-time PCR (qRT-PCR)

qRT-PCR was executed on a StepOnePLUS Real-Time PCR system (Applied Biosystems, USA) using 2×SG Green qPCR Mix with ROX (SinoGene, China, Cat. No. SG011) using primers QRT-BnaOASTL-F (5'-ACCCTGCCAACCCAAAGATA-3')

and QRT-BnaOASTL-R (5'-ACCACCAGTACCAATCCCAG-3'). *Actin* served as the internal control and amplified using Actin-F (5'-tttcctggcattgcagatcg-3') and Actin-R (5'-tccagacactgtacttgcgt-3'). Reactions (20 µL) included 2 µL cDNA, 0.4 µL of each primer (10 µM), 10 µL 2×SG Green Mix, and 7.2 µL ddH$_2$O. Cycling was run at 95°C for 3 min, then 40 cycles of 95°C for 10 s and 60°C for 30 s. Relative expression was quantified using the $2^{-\Delta\Delta Ct}$ method with three biological and three technical replicates.

## 2.10. Statistical analysis

All results are shown as mean ± standard deviation (SD) ≥3 independent replicates. Data were compared using one-way ANOVA with Tukey's HSD post hoc test (SPSS 22.0, IBM, USA), with $P < 0.05$ considered significant.

# 3. Results

## 3.1. Sequence and phylogenetic analysis of *BnaOASTL*

Gene structure analysis revealed that the *BnaOASTL* ORF is 969 bp long and encodes a 322-amino acid protein (S1 Fig). The mature protein has an estimated molecular weight of 33.9 kDa, a calculated pI of 5.50, and three conserved domains: TSGNT (substrate-binding site), KPGPHK (SAT1-binding site), and PXXSVKDR (PLP-binding site) (S2 Fig). Alignment with cysteine synthases from *Brassica rapa*, *Raphanus sativus*, *Hirschfeldia incana*, and *Eutrema salsugineum* showed amino acid sequence identities of 99.69%, 98.45%, 98.14%, and 97.52%, respectively. Phylogenetic analysis of the BnaOASTL protein with other 21 representative species (Fig 2) divided the 22 OASTL proteins into two subcategories. Subcategory I includes XP_049374434.1, NP_001274978.2, NP_001308271.1, XP_015088117.1, XP_059289749.1, KAJ8543230.1, KAF3681890.1, XP_012069631.1, XP_021593925.1, KAJ0233058.1, PWA95687.1 and GEV11055.1. Subcategory II includes XP_010532551.1, XP_020886714.1, XP_010488274.1, XP_006284124.1, NP_001190732.1, OASTL_NY868, O23733.1, KAJ0442914.1, XP_009148046.1 and CDY29778.1, with OASTL_NY868 closely related to O23733.1. Subcellular localization analysis indicated that BnaOASTL is cytoplasmic and nuclear, lacks transmembrane domains, and belongs to the PLP-dependent, β-substituted alanine synthase superfamily.

## 3.2. Subcellular localization of BnaOASTL

BnaOASTL-GFP was predominantly localized to the cytoplasm and nucleus (Fig 3), suggesting that BnaOASTL likely performs its functions in these subcellular compartments.

## 3.3. Molecular characterization of transgenic lines

Eight independent transgenic tobacco lines expressing the *BnaOASTL* gene were generated. Plants transformed with empty vector pBWA(V)HS-osgfp served as controls. Integration of the *BnaOASTL* in T3 plants was validated by PCR using gene-specific primers (Fig 4).

## 3.4. Phenotypic response to Cd stress

Tobacco plants were treated with 5 mg· kg$^{-1}$ Cd (CdCl$_2$·2.5 H$_2$O). After 7 days, transgenic and WT plants showed similar phenotypes under Cd stress, except for the slower growth of WT (Fig 5).

Panels a and g represent WT plants, and panels b–f and h–l represent transgenic lines.

## 3.5. BnaOASTL expression under Cd stress

qRT-PCR results (Fig 6) indicated that Cd stress significantly elevates *BnaOASTL* transcript abundance in roots and shoots, with levels in roots substantially exceeding those in shoots.

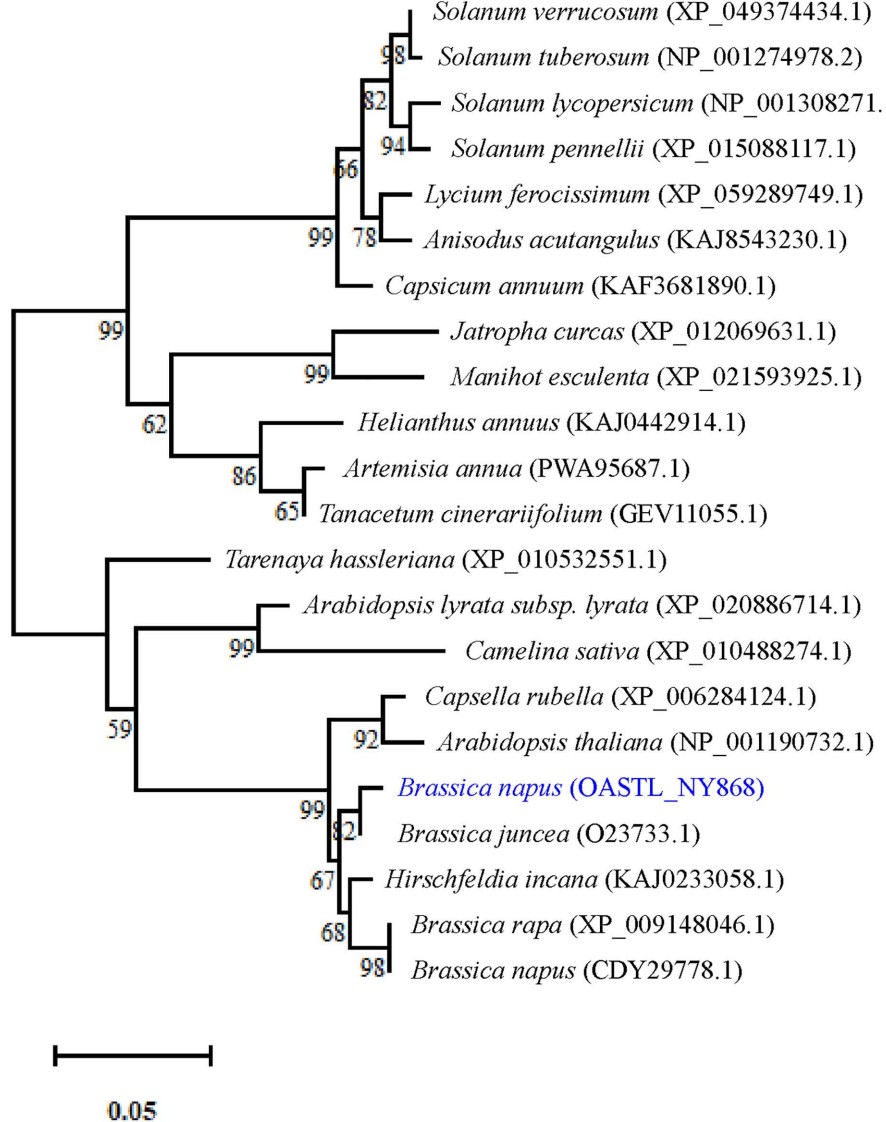

**Fig 2. Phylogenetic tree of *OASTL* proteins from *B. napus* and other species based on amino acid sequences.** Bootstrap values (>50%) from 1000 replicates are demonstrated at branch nodes. The scale bar represents a distance of 0.05. The *B. napus OASTL* sequence is marked with a blue symbol. Other sequences were obtained from XP_049374434.1, NP_001274978.2, NP_001308271.1, XP_015088117.1, XP_059289749.1, KAJ8543230.1, KAF3681890.1, PWA95687.1, GEV11055.1, KAJ0442914.1, XP_012069631.1, XP_021593925.1, OASTL_NY868, O23733.1, XP_009148046.1, CDY29778.1, KAJ0233058.1, XP_006284124.1, NP_001190732.1, XP_020886714.1, XP_010488274.1 and XP_010532551.1.

### 3.6. Cd and GSH contents

The expression of BnaOASTL in transgenic lines was further investigated by analyzing Cd contents in T3 transgenic and WT tobacco plants before and during Cd stress (Fig 7). Under both control and stress conditions, Cd levels were similar between *BnaOASTL* overexpression lines and wild-type plants. Similarly, GSH contents showed no significant differences between transgenic lines and WT under either normal or Cd-stress conditions (Fig 8).

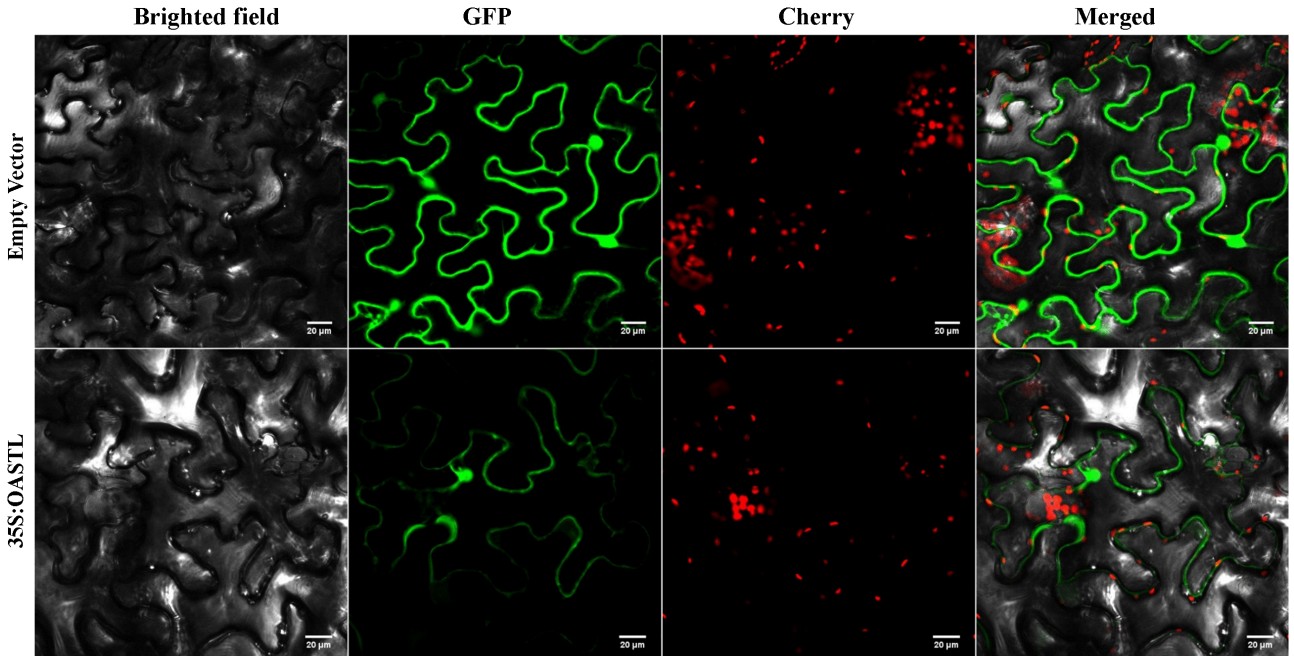

**Fig 3. Subcellular localization of BnaOASTL-GFP in tobacco epidermal cells.** Scale bar = 10 µm.

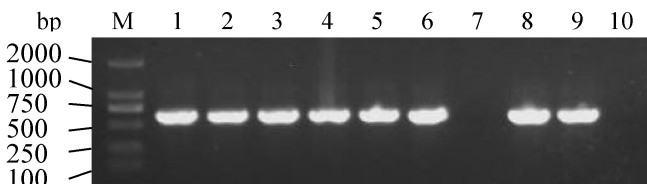

**Fig 4. Confirmation of *BnaOASTL* integration in tobacco lines by PCR, showing amplified *BnaOASTL* bands from 10 tobacco lines.** M, DL2000 DNA marker; 1–6 and 8: transgenic lines; 9, positive control (plasmid); and 10, negative control (WT).

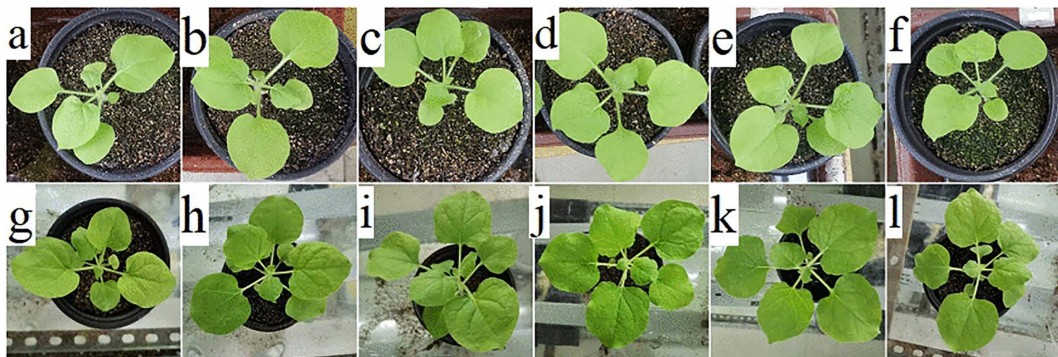

**Fig 5. Images of WT and transgenic plants before (a–f) and after (g–l) cadmium treatment.**

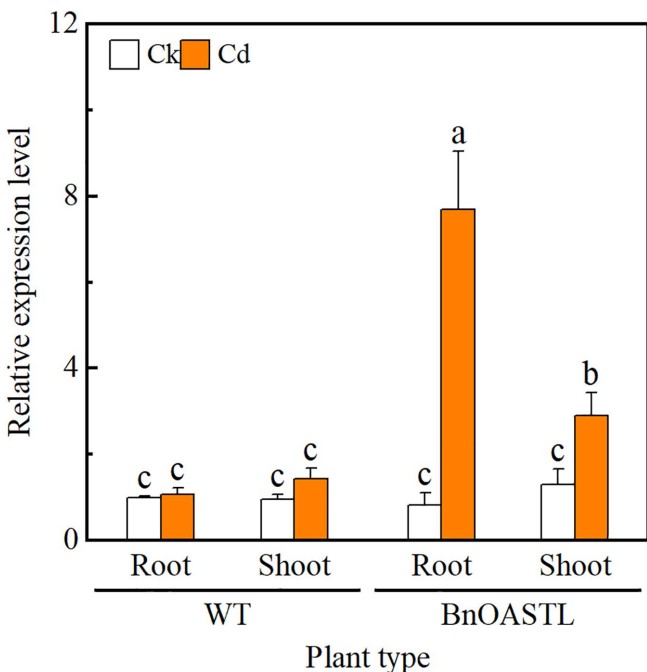

**Fig 6. *BnaOASTL* expression by qRT-PCR in roots and leaves of seedlings after 24 h treatment with 0 or 5 mg· kg⁻¹ Cd.** Data are shown as means ± SD (n = 3) with different letters indicating significant differences ($P < 0.05$, one-way ANOVA, Tukey's test).

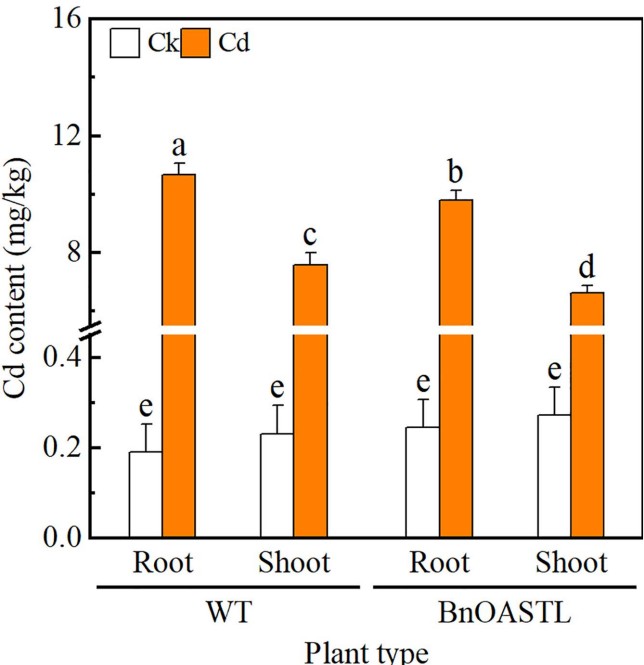

**Fig 7. Cadmium accumulation in shoots and roots of transgenic (T1-T4) and WT plants under 0 or 5 mg· kg⁻¹ Cd (CdCl₂·2.5 H₂O) treatment.** Values represent mean ± SD (n = 4), with different letters indicating significant differences ($P < 0.05$).

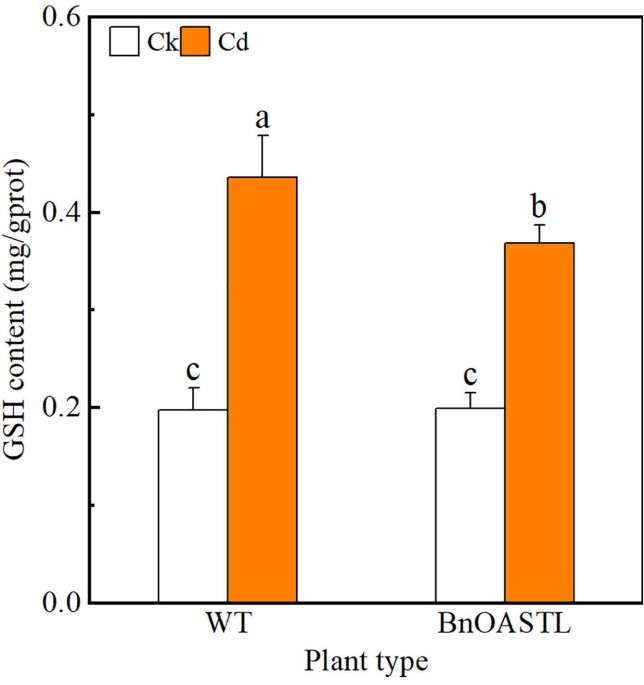

**Fig 8. Glutathione content in shoots.** Data are mean±SD; no significant differences were found.

## 4. Discussion

### 4.1. Sequence and structure analysis of *BnaOASTL*

Different OASTL isoenzymes perform diverse functions [16], and several family members contribute to plant responses to heavy metals and oxidative stress [5]. Recently, *OASTLs* have gained attention for their role in Cd stress responses [3]. In this study, a putative *OASTL* gene, designated *BnaOASTL*, was isolated from the high Cd-accumulating *B. napus* L. ('Nanyou 868'). Sequence analysis unveiled that *BnaOASTL* cDNA contains a 969 bp ORF encoding a 322-amino acid protein with a predicted MW of 33.9 kDa and a PI of 5.50. These features are consistent with OASTL proteins from *Arabidopsis*, Sorghum, and *Cardamine hupingshanensis*, which typically range from 305 to 433 amino acids and exhibit acidic values (<7) characteristic of cytoplasmic OASTLs [3,16]. The deduced BnaOASTL protein contains three conserved domains, i.e., TSGNT (substrate-binding site), KPGPHK (SAT1-binding site), and PXXSVKDR (PLP-binding site), indicating it likely uses pyridoxal monohydrate 5'-phosphate as a cofactor and belongs to the PLP-dependent β-substituted alanine synthase superfamily. The subcellular localization analysis predicts that BnaOASTL may exist in the cytoplasm, chloroplast, and mitochondrion. Confocal imaging of *BnaOASTL*-GFP in tobacco confirmed predominant localization in the cytoplasm and nucleus, consistent with previous reports highlighting cytoplasmic OASTL function [5,16]. These findings indicate that BnsOASTL belongs to the OASTLA1 type. Multiple sequence alignment unveiled highly conserved C-termini but more variable N-termini among OASTL proteins, suggesting both conserved function and potential for regulatory diversity. Phylogenetic analysis showed that BnaOASTL is most related to OASTL from *B. cretica* and *B. rapa*, confirming it as a genuine member of the OASTL family and representing a novel variant within this group.

### 4.2. Expression of *BnaOASTL* in transgenic lines in response to Cd

Morphological and physiological responses are key indicators of metal toxicity. To assess the functional role of *BnaOASTL* in Cd stress, transgenic tobacco overexpressing *BnaOASTL* was generated. Under Cd stress, no significant phenotypic

differences, such as chlorosis or leaf abscission, were observed between transgenic and wild-type seedlings, suggesting that tobacco inherently tolerates Cd at the applied concentration. qRT-PCR unveiled that *BnaOASTL* expression was dramatically induced in roots but only slightly in shoots of transgenic plants under Cd stress, consistent with patterns observed in other species [3,4]. This upregulation indicates that *BnaOASTL* is responsive to Cd stress and may participate in Cd-responsive pathways. However, despite the elevated transcript levels, transgenic plants did not exhibit enhanced Cd tolerance or accumulation compared to WT. Moreover, glutathione (GSH) content remained unchanged. These results suggest that overexpression of *BnaOASTL* alone is insufficient to alter Cd detoxification or accumulation in tobacco. This lack of phenotypic effect may be explained by several factors: (1) functional redundancy and compartmentalization of OASTL isoforms in different subcellular compartments (cytosol, chloroplasts, mitochondria), limiting changes in overall cysteine or GSH pools [15,18–20]; (2) post-translational regulation of OASTL via interaction with serine acetyltransferase (SAT) and feedback inhibition, which may prevent increased cysteine production without concomitant pathway activation; (3) species-specific context, as tobacco may lack regulatory elements or interacting partners present in *B. napus*, and (4) the complex of Cd detoxification networks, including phytochelatin synthesis, metal transporters, and the antioxidant systems [27], which cannot be significantly altered by overexpression of a single gene. These findings highlight the complexity of Cd stress responses and suggest that *BnaOASTL* likely functions within a broader metabolic and regulatory network rather than acting in isolation. However, these hypothesis needs a more in-depth study.

## 5. Conclusions

In summary, this study provides functional characterization of *BnaOASTL* from a high-Cd-accumulating rapeseed cultivar in a heterologous system. Although its overexpression in tobacco did not enhance Cd tolerance or accumulation, the gene's response to Cd stress underscores its involvement in Cd-responsive pathways. These findings highlight the complexity of Cd stress responses and suggest that *BnaOASTL* may function within a broader regulatory network.

## Supporting information

**S1 Fig. The complete cDNA sequence of *BnaOASTL* and its deduced protein sequence, with amino acids shown in one-letter code below the nucleotides and the asterisk denoting stop codon.**
(TIF)

**S2 Fig. Sequence alignment of BnaOASTL with OASTL proteins from various species.** Identical residues are marked with asterisks, conserved substitutions with colons, and semiconserved substitutions with periods. The conserved domains: TSGNT (substrate-binding site), KPGPHK (SAT1-binding site), and PXXSVKDR (PLP-binding site), are boxed. The BnaOASTL sequence was obtained from *B. napus*, and other sequences from XP_018470163.1, ACX70136.1, KAJ0233058.1, KAJ4871408.1, XP_006414628.1, XP_020874249.1, CAA58893.1, XP_006284124.1, and XP_009107806.1.
(TIF)

**S1 Raw image. Raw gel images.**
(PDF)

**S1 Data. Raw data for fig 6.**
(XLSX)

**S2 Data. Raw data for fig 7.**
(XLSX)

**S3 Data. Raw data for fig 8.**
(XLSX)

## Acknowledgments

The authors thank chatGPT for useful suggestions and editing the English language of the manuscript.

## Author contributions

**Data curation:** Jianwei Wang.

**Funding acquisition:** Xiaolan He.

**Writing – original draft:** Xiaolan He.

**Writing – review & editing:** Jianwei Wang, Wenxu Li, Xinhong Chen.

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
