## [Decision Letter · Decision Letter 0]

20 Aug 2025

Dear Dr. He,

Thank you for submitting your manuscript to PLOS ONE. After careful consideration, we feel that it has merit but does not fully meet PLOS ONE’s publication criteria as it currently stands. Therefore, we invite you to submit a revised version of the manuscript that addresses the points raised during the review process.

We look forward to receiving your revised manuscript.

Kind regards,

Naser A. Anjum, PhD

Academic Editor

PLOS ONE

Journal Requirements:

2. We noticed you have some minor occurrence of overlapping text with the following previous publication(s), which needs to be addressed: https://pubmed.ncbi.nlm.nih.gov/38775862/
https://www.mdpi.com/2311-7524/8/11/1002

In your revision ensure you cite all your sources (including your own works), and quote or rephrase any duplicated text outside the methods section. Further consideration is dependent on these concerns being addressed.

4. In this instance it seems there may be acceptable restrictions in place that prevent the public sharing of your minimal data. However, in line with our goal of ensuring long-term data availability to all interested researchers, PLOS’ Data Policy states that authors cannot be the sole named individuals responsible for ensuring data access (http://journals.plos.org/plosone/s/data-availability#loc-acceptable-data-sharing-methods).

5. Thank you for stating the following financial disclosure: [the Doctoral Development Special Project of Kaili University (BSFZ202503).].

6. Thank you for stating the following in the Acknowledgments Section of your manuscript: [Much appreciated financial support was provided by the Doctoral Development Special Project of Kaili University (BSFZ202503).]

Please remove any funding-related text from the manuscript and let us know how you would like to update your Funding Statement. Currently, your Funding Statement reads as follows: [the Doctoral Development Special Project of Kaili University (BSFZ202503).].

7. PLOS requires an ORCID iD for the corresponding author in Editorial Manager on papers submitted after December 6th, 2016. Please ensure that you have an ORCID iD and that it is validated in Editorial Manager. To do this, go to ‘Update my Information’ (in the upper left-hand corner of the main menu), and click on the Fetch/Validate link next to the ORCID field. This will take you to the ORCID site and allow you to create a new iD or authenticate a pre-existing iD in Editorial Manager.

8. PLOS ONE now requires that authors provide the original uncropped and unadjusted images underlying all blot or gel results reported in a submission’s figures or Supporting Information files. This policy and the journal’s other requirements for blot/gel reporting and figure preparation are described in detail at https://journals.plos.org/plosone/s/figures#loc-blot-and-gel-reporting-requirements and https://journals.plos.org/plosone/s/figures#loc-preparing-figures-from-image-files. When you submit your revised manuscript, please ensure that your figures adhere fully to these guidelines and provide the original underlying images for all blot or gel data reported in your submission. See the following link for instructions on providing the original image data: https://journals.plos.org/plosone/s/figures#loc-original-images-for-blots-and-gels.

Additional Editor Comments:

REVIEWER #1

The manuscript entitled "Physiological and Bn-OASTL Gene Expression Responses to Cadmium Stresses in Tobacco Seedlings" presents a systematic study on the cloning and heterologous expression of the BnOASTL gene and its role in cadmium stress response. The research question is relevant, and the methodology is generally sound. Although the expected enhancement of Cd resistance and accumulation was not observed, the negative findings provide valuable insight for future studies on heavy metal stress mechanisms. The manuscript is generally well-organized and provides sufficient data to support its conclusions.

However, some issues in the abstract, introduction, methods, results discussion, and language presentation need to be addressed to further improve the manuscript's clarity, scientific rigor, and academic value. I recommend a minor revision before acceptance.

1. The title, abstract, and keywords are not fully consistent regarding the research subject, species, and gene information. For example, the title and abstract refer to Bn-OASTL and tobacco, while the keywords include “rapeseed (Brassica napus L.)” instead. Please ensure that these elements are precisely aligned and consistently reflect the main experimental system and research focus.

2. The abstract contains some ambiguous statements and lacks clear logical flow. In particular, the functional analysis of the gene did not show a significant effect in transgenic tobacco, but the discussion on the underlying mechanisms is insufficient. Furthermore, the scientific significance and future perspective of this study in the context of Cd-tolerant crop molecular design are not adequately highlighted. Please refine the abstract to clearly state the main findings (e.g., the lack of improvement in Cd tolerance/accumulation) and emphasize their scientific relevance to the field.

3. The introduction is overly lengthy and the literature review is somewhat scattered. I recommend condensing the introduction to make it more concise, with a clear focus on the scientific question and the innovative aspects of this study. Excessive background information should be removed to avoid redundancy. Please clearly articulate the rationale for selecting BnOASTL, the reason for using tobacco as the heterologous expression system, and the current knowledge gaps that this work aims to address. In addition, the introduction should include a more forward-looking discussion on the potential of the OASTL gene family in molecular breeding and heavy metal phytoremediation, highlighting future directions and scientific significance.

4. Provide more experimental details, such as specific reagent sources, PCR/qPCR conditions, and statistical methods to improve reproducibility. Ensure all primers, vectors, and key reagents are clearly described.

5. Enhance the interpretation of the results, especially the possible reasons why overexpression of BnOASTL did not improve Cd tolerance. Discuss the broader significance and possible directions for future research, such as the need for multi-gene manipulation or systems biology approaches.

6. Ensure that all figures and legends are sufficiently detailed, and clearly indicate statistical significance where applicable.

7. Please ensure that all references are properly formatted and include DOI numbers where available. Including DOIs for all cited literature is required by the journal and will improve the accessibility and traceability of your references.

REVIEWER #2

The manuscript entitled "Physiological and Bn-OASTL Gene Expression Responses to

Cadmium Stresses in Tobacco Seedlings" deal with an original subject within the plant soil interactions field. neertheless the ms at its present form is not suitable for publication:

-the title did not much the Ms. the is no physiological study it is just a monitoring of Cd and GSH contents

-in the ms authors present a work on tobacco plants however in conclusion they change to tomato???

-molecular analyses are goods however the discussion is not appropriated to the founded results. it is clear from results that theBn-OASTL Gene has no relatioship with the resistance beahavior within tobacco plant meaning thet tobacco did not use the g O-acetylserine(thiol)lyase pathway to overcome the Cd stress.

Reviewers' comments:

Reviewer's Responses to Questions

**Comments to the Author**

1. Is the manuscript technically sound, and do the data support the conclusions?

Reviewer #1: Yes

Reviewer #2: Partly

2. Has the statistical analysis been performed appropriately and rigorously?

Reviewer #1: Yes

Reviewer #2: Yes

3. Have the authors made all data underlying the findings in their manuscript fully available?

Reviewer #1: Yes

Reviewer #2: Yes

4. Is the manuscript presented in an intelligible fashion and written in standard English?

Reviewer #1: Yes

Reviewer #2: Yes

Reviewer #1: The manuscript entitled "Physiological and Bn-OASTL Gene Expression Responses to Cadmium Stresses in Tobacco Seedlings" presents a systematic study on the cloning and heterologous expression of the BnOASTL gene and its role in cadmium stress response. The research question is relevant, and the methodology is generally sound. Although the expected enhancement of Cd resistance and accumulation was not observed, the negative findings provide valuable insight for future studies on heavy metal stress mechanisms. The manuscript is generally well-organized and provides sufficient data to support its conclusions.

However, some issues in the abstract, introduction, methods, results discussion, and language presentation need to be addressed to further improve the manuscript's clarity, scientific rigor, and academic value. I recommend a minor revision before acceptance.

1. The title, abstract, and keywords are not fully consistent regarding the research subject, species, and gene information. For example, the title and abstract refer to Bn-OASTL and tobacco, while the keywords include “rapeseed (Brassica napus L.)” instead. Please ensure that these elements are precisely aligned and consistently reflect the main experimental system and research focus.

2. The abstract contains some ambiguous statements and lacks clear logical flow. In particular, the functional analysis of the gene did not show a significant effect in transgenic tobacco, but the discussion on the underlying mechanisms is insufficient. Furthermore, the scientific significance and future perspective of this study in the context of Cd-tolerant crop molecular design are not adequately highlighted. Please refine the abstract to clearly state the main findings (e.g., the lack of improvement in Cd tolerance/accumulation) and emphasize their scientific relevance to the field.

3. The introduction is overly lengthy and the literature review is somewhat scattered. I recommend condensing the introduction to make it more concise, with a clear focus on the scientific question and the innovative aspects of this study. Excessive background information should be removed to avoid redundancy. Please clearly articulate the rationale for selecting BnOASTL, the reason for using tobacco as the heterologous expression system, and the current knowledge gaps that this work aims to address. In addition, the introduction should include a more forward-looking discussion on the potential of the OASTL gene family in molecular breeding and heavy metal phytoremediation, highlighting future directions and scientific significance.

4. Provide more experimental details, such as specific reagent sources, PCR/qPCR conditions, and statistical methods to improve reproducibility. Ensure all primers, vectors, and key reagents are clearly described.

5. Enhance the interpretation of the results, especially the possible reasons why overexpression of BnOASTL did not improve Cd tolerance. Discuss the broader significance and possible directions for future research, such as the need for multi-gene manipulation or systems biology approaches.

6. Ensure that all figures and legends are sufficiently detailed, and clearly indicate statistical significance where applicable.

7. Please ensure that all references are properly formatted and include DOI numbers where available. Including DOIs for all cited literature is required by the journal and will improve the accessibility and traceability of your references.

In summary, I recommend minor revision. The authors should carefully address the points above to enhance the scientific rigor, clarity, and academic value of the manuscript. After revision, the manuscript will be suitable for publication.

Reviewer #2: The manuscript entitled "Physiological and Bn-OASTL Gene Expression Responses to

Cadmium Stresses in Tobacco Seedlings" deal with an original subject within the plant soil interactions field. neertheless the ms at its present form is not suitable for publication:

-the title did not much the Ms. the is no physiological study it is just a monitoring of Cd and GSH contents

-in the ms authors present a work on tobacco plants however in conclusion they change to tomato???

-molecular analyses are goods however the discussion is not appropriated to the founded results. it is clear from results that theBn-OASTL Gene has no relatioship with the resistance beahavior within tobacco plant meaning thet tobacco did not use the g O-acetylserine(thiol)lyase pathway to overcome the cd stress

**Do you want your identity to be public for this peer review?** For information about this choice, including consent withdrawal, please see our Privacy Policy

Reviewer #1: No

Reviewer #2: No

---

## [Author Response · Author response to Decision Letter 1]

9 Oct 2025

We have carefully considered each comment and made corresponding corrections, which we hope will meet with your approval. All revisions are marked in red in the revised manuscript.

---

## [Decision Letter · Decision Letter 1]

14 Dec 2025

Monitoring of Cd and GSH Contents and Bn-OASTL Expression in Transgenic Tobacco Seedlings in Response to Cd Stress

PONE-D-25-39819R1

Dear Dr. He,

We’re pleased to inform you that your manuscript has been judged scientifically suitable for publication and will be formally accepted for publication once it meets all outstanding technical requirements.

Kind regards,

Debasis Chakrabarty

Academic Editor

PLOS One

Additional Editor Comments (optional):

Reviewers' comments:

Reviewer's Responses to Questions

**Comments to the Author**

Reviewer #1: (No Response)

Reviewer #3: All comments have been addressed

Reviewer #4: (No Response)

2. Is the manuscript technically sound, and do the data support the conclusions?

Reviewer #1: Yes

Reviewer #3: Yes

Reviewer #4: (No Response)

3. Has the statistical analysis been performed appropriately and rigorously?

Reviewer #1: Yes

Reviewer #3: Yes

Reviewer #4: (No Response)

4. Have the authors made all data underlying the findings in their manuscript fully available?

Reviewer #1: Yes

Reviewer #3: Yes

Reviewer #4: (No Response)

5. Is the manuscript presented in an intelligible fashion and written in standard English?

Reviewer #1: Yes

Reviewer #3: Yes

Reviewer #4: (No Response)

Reviewer #1: (No Response)

Reviewer #3: The MS titled "Monitoring of Cd and GSH Contents and Bn-OASTL Expression in Transgenic Tobacco Seedlings in Response to Cd Stress" has been well revised. And I have no additional comment.

Reviewer #4: (No Response)

**Do you want your identity to be public for this peer review?** For information about this choice, including consent withdrawal, please see our Privacy Policy

Reviewer #1: **Yes:**  Hucheng XING

Reviewer #3: No

Reviewer #4: No

---

## [Editor Report · Acceptance letter]

20 Aug 2025

PONE-D-25-39819R1

PLOS One

Dear Dr. He,

I'm pleased to inform you that your manuscript has been deemed suitable for publication in PLOS One. Congratulations! Your manuscript is now being handed over to our production team.

Kind regards,

on behalf of

Dr. Debasis Chakrabarty

Academic Editor

PLOS One